# Neural Foundations of Ayres Sensory Integration^®^

**DOI:** 10.3390/brainsci9070153

**Published:** 2019-06-28

**Authors:** Shelly J. Lane, Zoe Mailloux, Sarah Schoen, Anita Bundy, Teresa A. May-Benson, L. Diane Parham, Susanne Smith Roley, Roseann C. Schaaf

**Affiliations:** 1Department of Occupational Therapy, College of Health and Human Sciences, Colorado State University, Fort Collins, CO 80523, USA; 2Discipline of Occupational Therapy, School of Health Sciences, University of Newcastle, Newcastle, New South Wales 2300, Australia; 3Department of Occupational Therapy and Farber Institute for Neuroscience, Thomas Jefferson University, Philadelphia, PA 19107, USA; 4STAR Institute for Sensory Processing Disorder, Greenwood Village, CO 80111, USA; 5The Spiral Foundation, Newton, MA 02458, USA; 6Department of Pediatrics, Occupational Therapy Graduate Program, University of New Mexico, Albuquerque, NM 871321, USA; 7Collaborative Leadership in Ayres Sensory Integration, Redondo Beach, CA 90277, USA

**Keywords:** sensory integration, neuroscience, sensory registration, sensory modulation, sensory processing, sensory perception, sensory reactivity, dyspraxia

## Abstract

Sensory integration, now trademarked as Ayres Sensory Integration^®^ or ASI, is based on principles of neuroscience and provides a framework for understanding the contributions of the sensory and motor foundations of human behavior. The theory and practice of ASI continues to evolve as greater understanding of the neurobiology of human behavior emerges. In this paper we examine core constructs of ASI identified in the seminal work of Dr. Jean Ayres, and present current neuroscience research that underlies the main patterns of sensory integration function and dysfunction. We consider how current research verifies and clarifies Ayres’ propositions by describing functions of the vestibular, proprioceptive, and tactile sensory systems, and exploring their relationships to ocular, postural, bilateral integration, praxis, and sensory modulation. We close by proposing neuroplasticity as the mechanisms underlying change as a result of ASI intervention.

## 1. Introduction

A. Jean Ayres (18 July 1920–16 December 1988) was an occupational therapist and neuropsychologist who spent her career conducting research and developing theory and intervention strategies to help her understand and treat children with learning and behavioral challenges. Ayres relied heavily on neuroscience literature to guide her understanding of previously-unexamined sensory (and motor) deficits affecting learning and behavior. Studying with Margaret Rood, Ayres’ early research examined the use of proprioception to facilitate voluntary movement needed for everyday activities [1,2,3]. In the 1970s, Ayres began to publish work describing difficulties processing and integrating sensation that occurred in some children with learning disorders. Based on her research and clinical experience, she developed the theory and practice of sensory integration which describes how the nervous system translates sensory information into action and posits that adequate sensory integration is an important foundation for adaptive behavior [4,5]. Sensory integration theory emphasizes the active, dynamic sensory–motor processes that support movement as well as interaction within social and physical environments and that act as a catalyst for development. Ayres’ theory and practice emanated from a decades-long program of research. Today this body of work is recognized as Ayres Sensory Integration^®^ (ASI) and includes theory, postulates about the mechanisms of sensory integration’s effects, assessment strategies to identify challenges in sensory integration, intervention principles, a manualized intervention to guide treatment, and a measure of fidelity that is used to support research and practice [6,7,8,9]. 

Acting only with knowledge of neuroscience at the time in which she lived, Ayres was prescient in her thinking. Much of her work remains supported by current literature. Ayres was guided by two important principles: “the brain is a self-organizing system” and “intersensory integration is foundational to function”. Typically, she began papers or lectures with simple hypotheses reflecting these principles. An example: “The brain, under normal circumstances, is a self-organizing system. When it is unsuccessful in accomplishing its integrative task, the behavior directed by the brain fails to fall within ‘normal’ expectations”, [10] (p. 41). Ayres then continued, eloquently, supporting her hypotheses with neuroscience. For example:
“*Intersensory integration can occur within a single neuron, a nucleus or the diencephalon, an entire hemisphere or even between hemispheres. One of the methods by which the CNS [Central Nervous System] integrates sensory information from several different sources is by directing it to a common neuron called a convergent neuron. Whenever there is multiplicity of input all related to a single sensorimotor process, there is probably convergence of input.*”[10] (p. 42).

Neuroscience remains a cornerstone of the theory today, now reflecting contemporary techniques [6,11,12]. For example, in relation to autism spectrum disorders (ASD), Kilroy, Aziz-Zadeh and Cermak [11] recently explored postulates proposed by Ayres (e.g., registration, modulation) in the context of neuroimaging literature, focusing on the substantial body of research that examines sensory integrative dysfunction in children with ASD. 

In this paper we examine core constructs of ASI, as articulated by Ayres, reflected in the context of contemporary neuroscience. Specifically, we consider the neuroscience foundations of ASI in the areas of sensory perception (vestibular, proprioceptive, and tactile systems), the relationship of these to ocular, postural, bilateral integration, and praxis functions, and the construct of sensory modulation. We close by discussing neuroplasticity as the key mechanism of lasting neural change as a result of ASI intervention. Throughout, we consider how current research verifies and clarifies Ayres’ propositions.

## 2. A Foundation in Sensory Systems

### 2.1. Vestibular System: Function and Impact

Vestibular receptors are formed early in fetal development and are functioning at birth [13]. This system contains two receptor types: the semicircular canals that detect angular movement of the head and the otolith organs (utricle and saccule) that detect linear movement, and the pull of gravity. Consequently, the vestibular system provides the brain with critical information regarding velocity and direction of head movement and static head position relative to gravity. This system operates continuously and unconsciously in the background of everyday life.

Early on, Ayres [4] proposed that the vestibular system had a significant influence on brain and behavior functions. Research now supports her original thinking, showing that vestibular information travels to many brain structures that serve a number of critical functions: arousal regulation, static and dynamic postural control, balance and equilibrium responses, bilateral coordination, maintenance of a stable visual field, and spatial perception for efficient navigation of the body through space [13,14,15,16]. For example, rapid or unpredictable acceleration of the body through space, such as occurs during a rollercoaster ride, is associated with increased alertness (e.g., excitement) and autonomic responses that activate the arousal system via the brainstem reticular formation [17]. Slow, rhythmic movement, such as rocking, produces the opposite effect: decreased arousal experienced as calming or drowsiness [18]. Vestibular inputs carried via the vestibulospinal tracts selectively activate neck and trunk musculature for effective postural and head control, whether the person is stable or moving [19]. In addition, vestibular information travels from the brainstem to the cerebellum to ensure refined and efficient postural and head control [20]. The sum of these connections is crucial for effective postural control. Furthermore, they lay a foundation upon which complex motor skills can be developed. 

Vestibular information, carried in the medial longitudinal fasciculus to the cranial nerves controlling the extraocular muscles, also supports coordinated eye and head movements. These connections enable the eye muscles to rapidly and precisely adjust eye positions while the head is moving, so that a moving person nonetheless perceives the visual surround as perfectly stable—even when shifting gaze during movement [21]. Through strong connections with vision and proprioception, the vestibular system contributes to anticipatory motor actions and plans. Further, because the vestibular system is a bilateral system that affects muscle activation throughout the body, it contributes to bilateral motor coordination. In fact, both Ayres [22,23,24] and Mailloux et al. [25] found that decreased vestibular responses during her postrotary nystagmus test were associated with decreased bilateral integration, decreased extension against gravity (i.e., prone extension posture), poor equilibrium reactions and decreased coordination of eye and head movements. She termed this pattern “deficits in vestibular-bilateral integration” [24].

Widespread vestibular influences on behavior and attention led Ayres to examine whether the vestibular system might play an important role in learning and behavioral difficulties of children. She hypothesized that vestibular processing inefficiencies would negatively affect the higher-level cognitive functions needed for academic learning, as well as the arousal regulation capacities needed for self-regulation of emotions and behavior. She further speculated that the vestibular system could be engaged in the context of individualized intervention to help children function more fully and successfully in school, at home, and in play [26]. During Ayres’ lifetime, the idea that vestibular processing contributed to high-level cortical functioning was considered inappropriate by many scientists because they believed that vestibular projections did not reach the cerebral cortex. However, even in the 1960s researchers identified cortical areas that received vestibular input [27]. More recently, neuroscientists have shown that a number of cortical areas (including the temporo-parietal junction, anterior parietal lobe, posterior parietal and medial superior temporal cortices, cingulate gyrus and retrosplenial cortex, and hippocampal and parahippocampal cortices) all receive vestibular information [28]. Additionally, researchers have shown that the vestibular system plays an important role, not only in spatial memory, but also in object recognition and numerical cognition [28]. Thus, contemporary neuroscience provides support for Ayres’ early thinking.

### 2.2. Somatosensory Function and Impact

Somatosensation comprises touch and proprioception, two systems that develop very early. A fetus responds to tactile input at 7–8 weeks [29] and proprioception by 10–12 weeks [30]. Researchers have long understood that tactile sensations project to the primary sensory cortex (S1) and the secondary somatosensory cortex (S2), a region that contributes to object manipulation and grasp as well as to tactile discrimination. Information sent to S2, in turn, plays a role in linking present and past sensations, a function that is crucial to motor planning [31].

As Ayres hypothesized, ties between somatosensation and other sensory systems are strong. Tactile sensations project to the posterior parietal cortex where they are integrated with visual information and motor signals [32]. Research also unequivocally shows that somatosensory–vestibular–visual integration occurs at multiple CNS levels: the vestibular nuclei, the thalamus, and the cortex [33]. Integration of these multisensory inputs is foundational to detection of self-motion, postural stability, and spatial orientation [33,34,35,36,37,38]. Ferre and colleagues [39] also indicated that vestibular activation increases tactile discrimination and may decrease pain. Tactile input to the insula also plays a role in homeostatic regulation and interoception [40], and projects to the orbitofrontal cortex, and contributes to affect [41].

The knowledge that touch serves multiple functions in the body, ranging from simple (reflex withdrawal from a painful stimulus) to complex (stress reduction associated with massage and integration with other sensations), led Ayres to hypothesize that tactile inputs had a pervasive influence on central nervous system processes [4]. She indicated that integration of touch with other sensations was evident and important from a very early age [4]. Recent research provides ample support that this is the case. For example, Addabbo and colleagues [42] found that newborns were more apt to shift gaze toward touch when the touch was from another person as opposed to from an object.

Ayres was so convinced of the importance of somatosensory processing that she concluded, “in the child up to eight or nine years of age the degree of integration of the tactile system is a reasonably accurate—but not invariable—index of sensory integration in general,” [4] (p. 62). She further hypothesized that the somatosensory system plays a crucial role in praxis. This hypothesis was grounded in repeated studies showing clear relationships among tests of tactile perception and praxis, conducted using factor analysis and regression analysis [22,24,25,43,44,45,46,47]. Recent science supports these findings showing that somatosensation impacts feedforward processing, and prediction of movement [31].

## 3. Sensation Informing Action: Praxis

Ayres [48,49] defined praxis as “the basis for dealing with the physical environment in an adaptive way… dressing, eating with utensils, playing, writing, building, driving an automobile, changing the physical environment to meet a purposeful goal, and making a living” [49] (p. 44).

Praxis is a core construct of ASI and comprises three components: ideation (conceptualization of actions), motor planning, and execution. As evident in her earliest work [50], Ayres always emphasized the key role of visual [51] and somatosensory [52] perception in praxis. Using factor and cluster analyses with Sensory Integration and Praxis Test data [45] (SIPT) Ayres found three patterns of praxis dysfunction; she labelled these somatodyspraxia, visuodyspraxia, and dyspraxia on verbal command.

As early as 1961 and 1962, [52,53], Ayres emphasized body schema as the foundation for praxis and its relationship to somatosensation. Based on this hypothesized relationship, she identified key tenets of intervention for dyspraxia: engagement in active, sensory–motor activities that present the just-right challenge [48,53] and elicit adaptive motor responses [10]. These tenets remain current today.

Numerous researchers have built on Ayres’ studies to understand the nature of praxis. Mulligan [46] conducted a large-scale (*N* > 10,000) confirmatory factor and cluster analyses of SIPT data and confirmed Ayres’ findings of a somatosensory-based dyspraxia and bilateral integration and sequencing dyspraxia. More recently Mailloux et al. [25], Van Jaarsveld et al. [47], and Smith Roley et al. [54] also confirmed the patterns identified by Ayres in retrospective factor analyses.

While Ayres’ [48] described ideation or concept formation as a key component of praxis, she did not actively assess ideational praxis. More recently, however, May-Benson and Cermak [55] delved more deeply into ideational praxis, developing an assessment that has allowed them to evaluate and expand the ideational aspect of praxis. In a group of 60 children, May-Benson [56] distinguished ideational dyspraxia from other disorders of praxis.

Because information on the neurological underpinnings of motor performance was limited, especially in children, Ayres [48] relied on the literature on adult apraxia. Today, however, we are able to apply contemporary neurological research on motor performance and motor planning to praxis and dyspraxia [57]. Neuroimaging techniques, for example, have confirmed many of the connections among sensation, motor performance and praxis that Ayres proposed originally. While there are limited neuroscience studies specifically on dyspraxia, many researchers have studied children with developmental coordination disorder, autism, and other similar motor coordination problems. Many link ideation and motor by focusing on the relationship of visual perception, conceptualization, and motor planning. Vianio et al. [58] proposed that the conceptualization of object use in planned motor actions forms in cells in the anterior intraparietal sulcus of parietal cortex (AIP). Cells in the AIP appear to extract object-specific perceptual information, relevant to a particular motor action, from the visual stream and send those stimuli on to area F5 and the ventral premotor cortex. Using a computer-generated visual display, O’Brien et al. [59] found that children with dyspraxia have greater difficulties with form and motion coherence than children with other developmental delays or typically-developing peers. More recently in fMRI (functional magnetic resonance imaging) studies with 32 young adults Zapparoli et al. [60] found that internally driven intentional actions, such as those utilized in motor planning, were associated with activation of premotor and prefrontal areas: pre-supplemental motor area, supplemental motor area (SMA), and angular cingulate cortex (ACC). Finding atypical activation in frontal, parietal, and cerebellar areas in adults with motor coordination difficulties, Kashuk et al. [61] proposed that motor coordination impairments may be associated with disruption of parieto–frontal and parieto–cerebellar networks. Other researchers have uncovered new information on the relationship of tactile processing to motor performance. Cox et al. [62] found that 20 children with developmental coordination disorder who had decreased responses to localized touch to the hands also had poorer handwriting and decreased speed of functional hand skills. Similarly, using diffusion tensor imaging (DTI) with a group of 60 adults with ASD, Thompson et al. [63] found that decreased sensory processing in individuals with ASD was related to decreased motor performance and that direct interaction between the primary sensory cortex (S1) and the primary motor cortex (M1) may contribute to the ability to precisely interact with, and manipulate, the environment. These studies demonstrating the sensorimotor basis of praxis confirm Ayres’ original conceptualizations of the neuroscience underpinning praxis.

## 4. Modulating Sensory Responses

Sensory modulation disorders comprise exaggerated (either hyper-reactive or hypo-reactive) responses to sensation. These interfere with engagement in daily activities such as eating, grooming, and socializing [4]. Ayres began her research, and thus her theory development, with a desire to understand how the sensory systems functioned, both independently and in integrated ways. While she initially considered each sensory system in a unified fashion (e.g., “tactile function” or “vestibular function”), she consistently found a need to distinguish sensory perception from sensory modulation. While sensory perception in any system informs planned action and cognition, sensory modulation serves a more regulatory function. Notably, Ayres consistently associated somatosensory perceptual functions with praxis abilities, as noted above, and sensory modulation with attention, arousal, activity level, and emotion regulation. She also repeatedly identified two conditions that reflect poor modulation of tactile and vestibular sensations. In early studies, she linked over-reactivity to tactile input with behavioral hyperactivity and distractibility [64]. She named this condition “tactile defensiveness” [43,65]. Later she associated disproportional fear of movement with over-reactivity to vestibular input, naming this condition “gravitational insecurity” [66,67].

As the prevalence of autism increased toward the end of her career, Ayres [66] noted heightened sensitivity to auditory, visual, olfactory, and gustatory sensations in this population. These hyperreactive responses commonly manifest as distress and lack of habituation to auditory and tactile stimuli. Conversely, she documented observations that suggested a failure to notice or register stimuli that would be salient to most children and termed this “sensory registration difficulties.” Children with poor registration might fail to orient to a sound or visual stimulus that most children would notice or behave as though a tactile stimulus never occurred.

Ayres’ observations of behaviors suggesting poor sensory modulation provided a foundation for other researchers who subsequently examined potential neurophysiological underpinnings. These investigations included examinations of autonomic nervous system responses to sensation, the role of arousal in the process of attention allocation, and integration among the sensory systems in the service of functions such as self-regulation and focusing and shifting of attention. Here we focus on three lines of research related to sensory modulation deficits: (1) arousal mechanisms and the autonomic nervous system, (2) brain mechanisms related to sensory filtering, and (3) brain regions related to multisensory integration.

Physiological mechanisms associated with the autonomic nervous system were first investigated because individuals with sensory hyperreactivity displayed an unusually strong fight or flight response in the presence of aversive, and often non-aversive, sensory experiences. Miller and colleagues [68] observed increased activation of the sympathetic nervous system in a research paradigm called the Sensory Challenge Protocol. Within several diagnostic groups, individuals with hyperreactivity to sensation had elevated electrodermal activity (a measure of sympathetic activity) and slower habituation in response to repeated sensory events across five sensory domains [68,69,70,71,72]. Lane and colleagues [73] identified the magnitude of electrodermal response to sensory challenge as a mediator between high levels of alertness and behavioral anxiety and sympathetic recovery from a sensory challenge. Additionally, Schaaf et al. [74] documented inadequate parasympathetic activation (i.e., lower vagal tone) in children with sensory hyperreactivity compared to controls. These investigators interpreted this as reflecting autonomic imbalance, with increased parasympathetic activation to regulate the sympathetic system and subsequent reactivity to sensation.

The ability to filter out redundant or unnecessary stimuli has also been hypothesized as an underlying deficit of individuals with poor sensory modulation (i.e., hyper-reactivity), which researchers interpret as evidence of over-processing of low-salience stimuli. High density electrophysiology recordings during a sensory gating paradigm suggested that both adults [75] and children [76] with sensory hyper-reactivity had less efficient sensory gating than their typically-developing counterparts.

Multi-sensory experiences are part of everyday life and play important roles in development. In typically developing individuals, multisensory inputs enhance processing and interpretation of sensory events [77]; concordant sensory stimuli result in enhanced evoked related potentials. In contrast, individuals with poor modulation (i.e., hyperreactivity) show weaker event-related potentials to auditory and somatosensory stimuli, individually, and atypical auditory–somatosensory integration, causing them to become overwhelmed in situations where such multiple modality sensations are present [78]. This has been shown in children [78,79] and adults [80,81,82].

Evidence from diffusion tensor imagining (DTI) studies further confirm that impairments in multisensory integration may be related to sensory hyper-reactivity. Brain images of children with sensory hyper-reactivity identified by the Sensory Profile revealed reduced white matter microstructure in parietal and occipital tracts, where integration of auditory, tactile, and visual information occurs [83,84]. Although these primary sensory processing tracts were affected in both children with ASD and those with hyper-reactivity but no other clinical diagnoses, children in the latter group tended toward lower connectivity compared to children with ASD suggesting that the connectivity differences reflected sensory hyper-reactivity rather than being a marker of ASD. Children with ASD had distinct differences in regions of the amygdala and hippocampus associated with social emotional processing [84]. In a separate study, children with Attention Deficit Hyperactivity Disorder showed decreased connectivity in tracts of the prefrontal region that mediate motor, cognitive, and behaviors functions [85,86].

Using fMRI Green and colleagues [87] also found atypical connectivity and multi-sensory integration associated with sensory hyper-reactivity. This work examined the salience network, made up of a number of structures (i.e., anterior insula and other insular regions, anterior cingulate cortex, temporal poles, dorsolateral prefrontal cortex, amygdala) supports our ability to decide which environmental sensory input has meaning (salience) in the moment, warranting attention. In individuals with ASD and sensory hyper-reactivity greater connectivity within the salience network, and between this and primary sensory cortex, was identified along with reduced connectivity to visual association areas. A similar pattern was found in individuals without ASD. These atypical connectivity findings suggest that sensory hyper-reactivity itself is linked to allocating too much attention to basic external sensory input, while also limiting attention to social cues.

Poor sensory modulation manifesting as hypo-reactivity has received some attention in the literature, although it is more challenging to study because the behavioral indicators are less clear. Hypo-reactivity has been identified in the auditory domain in individuals with normal hearing capabilities. In fact, researchers using the Sensory Profile to measure sensory reactivity have found that 50–75% of children with ASD show hypo-reactivity to auditory stimuli [88,89]. Further, several researchers have associated hypo-reactivity with poor academic performance and social functioning [90,91]. Interestingly, in rodents, auditory hypo-reactivity is associated with altered neural connectivity and morphological changes in the subcortical auditory system [92]. These findings suggest potential neural mechanisms underlying deficits in performance associated with hypo-reactivity to sensation.

In summary, a number of researchers have examined potential neural underpinnings of sensory hypo- and hyper-reactivity. This rich area of research will continue to contribute to deeper understanding of modulation disorders and more informed approaches to intervention.

## 5. Sensory Integration: Influencing Neuroplasticity

ASI intervention is based on the concept of neuroplasticity, that is, that the nervous system changes in response to experience. Thus, through guided participation in sensorimotor activities targeting a child’s individual needs, ASI intervention is hypothesized to improve function, skill, and behavior as a basis for participation in everyday activities. More specifically, ASI proposes that active engagement, in individually-tailored sensorimotor activities, contextualized in play, at the just-right-challenge, promotes adaptive behaviors via neuroplastic changes that occur in response to these experiences.

The idea that the brain can change in response to experience is supported by neuroscience research that shows that experience-dependent learning shapes both brain function and behavior. Experience-dependent learning is the ongoing process of creation and organization of neuronal connections that occur as a result of experiences. An understanding of experience-dependent learning emerged from early experiments by Donald Hebb [93] who showed, in simple organisms, that synaptic efficacy improves when presynaptic cells repeatedly and persistently stimulate the postsynaptic cell. In other words, neurons that fire together, wire together, a statement that reflects Hebbian learning. This principle underscores the notion that neurons or neuron pools respond to stimulation and resultant output in adaptive ways—they begin to function as an organized unit of activity firing together more readily to produce output. This concept provided an important foundation for many studies of neuroplasticity and enrichment. For example, Diamond and colleagues [94], and Markham and Greenough [95] showed that rodents who were housed in enriched environments and given the opportunity to actively explore and engage with the toys and objects evidenced conformational changes in brain structure and organization. These studies showed that the brain continues to form new synaptic connections throughout life in response to environmental enrichment and learning, solidifying our knowledge of neuroplasticity. Subsequent studies targeting neuroplasticity and enriched environments further demonstrated the brain’s neuroplastic abilities; it is now widely accepted that the brain changes its structure and function in response to enrichment. Changing the input by providing novel experiences and learning opportunities leads to brain changes that include increased gray matter density, angiogenesis, glial volume, and neurogenesis [96]. Reynolds, Lane, and Richards [97] reviewed environmental enrichment studies with animals, finding that participation in enriched environments with novel sensory, motor, and cognitive challenges “can initiate lasting functional changes in the brain,” (p. 129). These investigators concluded that several key principles of the enriched environment principles are a match with ASI intervention, lending support to the premise that ASI promotes neuroplasticity. Ayres’ early work applied concepts of neuroplasticity to intervention and underscored the role of neuroplasticity in creating changes in function and behavior via the individually-tailored sensorimotor activities, contextualized in play at the just-right-challenge to facilitate adaptive behaviors: the hallmark of ASI.

Subsequent studies with children offer evidence in support of ASI promoting change in function and participation. Foundational to our ability to study the outcomes of ASI intervention, Schaaf and Mailloux [8] operationalized the principles of ASI into a manualized protocol. This manualized protocol guides (1) clinical reasoning around the sensory and motor factors impacting participation in everyday activities and (2) design of multisensory activities targeting these sensorimotor mechanisms. Accordingly, the intervention provides “enriched experiences” to facilitate neuroplasticity as a basis for improved function and participation in daily activities. Schaaf and colleagues’ [98] participants with ASD who received ASI intervention scored significantly higher on functional skills and participation in daily activities measured by Goal Attainment Scales (GAS) (*p* = 0.003, *d* = 1.2) and showed significantly greater improvements than a control group matched for IQ and ASD severity in independence in self-care (*p* = 0.008, *d* = 0.9) and socialization (*p* = 0.04, *d* = 0.7) on the Pediatric Evaluation of Disability Inventory [99]. These researchers ensured treatment fidelity through a manualized protocol and a validated ASI Fidelity Measure [7,100]. In an earlier randomized control study, Pfeiffer and colleagues [101] examined ASI intervention relative to a fine motor intervention in a summer camp environment for children with ASD. These investigators also ensured rigor using manualized interventions and fidelity tools for both groups. Results showed that children with ASD receiving ASI improved more on their GAS goals than a comparison group.

Based on the results of Schaaf and colleagues’ and Pfeiffer and colleagues’ studies, it has been determined that ASI meets the criteria for an evidence-based intervention for children with ASD, improving functional skills for participation in everyday activity [102,103]. These collective findings suggest that improvements are at least partially related to neuroplasticity. Schaaf’s team is now studying this premise by measuring change post-intervention, via an evoked potential paradigm created by Foxe and colleagues [80,104]. These data will determine whether changes in function are accompanied by concomitant changes in the brain’s ability to manage multisensory information; and whether improvements are sustained after three months. If so, this will be the first evidence that ASI does, as Ayres suggested, improve function via neuroplasticity.

## 6. Discussion

This paper reviews the neural foundations of sensory integration and praxis that inform Ayres Sensory Integration^®^ (ASI) theory, as well as the neuroplasticity principles that guide ASI intervention. We examine the historic and current neuroscientific research that is relevant to the main patterns of sensory integration disorders, with special attention to the primary functions of the vestibular, proprioceptive, and tactile sensory systems, and their relationships to ocular control, postural stability, bilateral integration, praxis, and sensory reactivity. This knowledge base is particularly relevant to child development and participation in everyday life activities, as it points to the critical role of sensory–motor experiences for cognitive development, movement skills, emotion regulation, social relationships, and activity participation in early childhood [5,105]. It is also highly relevant to therapeutic practices in assessment of foundational sensory integration and praxis abilities, as well as intervention that targets deficiencies in sensory integration and praxis across the lifespan.

Knowledge emanating from the neuroscience of sensory integration should inform not only individualized interventions for infants and children, but also the design of contemporary health and education systems. Preschool children today, even those living in affluent areas, generally display poor school readiness [106]. Older children are expected to sit for longer and longer periods at school, with recess greatly reduced or eliminated. Reduction or elimination of recess, which provides children with opportunities for sensory–motor exploration and skill development, has long been shown to be detrimental to learning. Additionally, a body of research shows that frequent classroom breaks with movement experiences result in improved academic performance [107,108].

Further, prevalence rates are growing for diagnoses that characteristically involve sensory integration and praxis difficulties. For example, diagnoses of attention deficit disorders with and without hyperactivity are on the rise [109], and at least one study indicates that vestibular processing challenges, previously unidentified in this population, may affect some children with this diagnosis [110]. Sensory integration and praxis concerns, including atypical reactivity, are prevalent among children with autism [45,54]. This is a population with epidemic growth, as reflected in dramatic changes in prevalence estimates reported Center for Disease Control (CDC): 1 in 166 in 2004, and 1 in 59 in 2014 [111]. In addition, many low incidence populations also are characterized by one or more sensory integrative deficit, such as poor vestibular functions in children with cochlear implants [112] and multiple sensory processing deficits among premature infants, which are not outgrown in childhood [113,114,115].

These growing concerns suggest a call to action for more sophisticated knowledge of the complex neural processes implicated in sensory integration and praxis difficulties. This knowledge is essential in order to develop effective, individually tailored interventions for children affected by these challenges. Further, this knowledge may contribute to the development of system-wide designs for schools, health care facilities, and community spaces that support sensory integration and praxis for people of all ages.

## 7. Conclusions

Advances in neuroscience over the last several decades have allowed contemporary scientists to confirm and clarify some of the sensory integration and praxis patterns of sensory–motor functioning that emerged from Ayres’ research on children with learning and behavioral difficulties. Contemporary imaging tools, as well as ongoing research on learning, attention, and behavior, will enable neuroscientists today to further refine and expand on Ayres’ work. As a body of research and practice, Ayres Sensory Integration^®^ reflects an effort to understand the human condition in a meaningful way, contributing to the ways that we understand how differences in neural functioning affect participation and engagement.

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
