# Peer review of "Neural Foundations of Ayres Sensory Integration®"

_brainsci, 2019, doi:10.3390/brainsci9070153_

Round 1
Reviewer 1 Report
Excellent review paper.
I find the submitted manuscript as an excellent and rather comprehensive review of the neural foundation. The authors provided a very balanced paper and linked together a range of issues and studies spanning from behavior to neuronal level, from psychological evaluations to most recent functional brain imaging studies. The general topic of neural foundation of the sensory integration is very much open and underexplored area of research with extremely relevant implications for a range of developmental problems, therapies, education,... I don't have any specific additional content suggestions for the authors of this review paper. Their coverage of the existing ASI relevant literature regarding a range of functional brain imaging studies that explored sensory integration is rather minimal and focusing mostly on EEG studies but I find that acceptable in the context of the scope of their review.
Minor changes:
line329: missing page number for citation from ref. 96
lines 345 and 367: no need to redefine acronyms ASD and ASI, respectively
lines 595 and 661: sam reference included twice, first as ref. 80 and later as ref. 104
Author Response
Thank you for your careful review of this manuscript. The in-text edits have been addressed, and the references have been changed. The duplicate of reference 80 as reference 104 has been corrected and subsequent reference numbers changed accordingly.
Reviewer 2 Report
This is a well written and well-organized paper that clarifies the core constructs of Ayres Sensory Integration (ASI) and the current research that verifies and clarifies Ayres’ SI work. This paper makes an important contribution to our current understanding of the underlying mechanisms contributing to sensory integration, ASI, and implications for other interventions. A few minor edits are suggested.
1. Line 60, check spacing after “today.
2. Line 231, change “thought” to “though.”
3. Line 329, need to add page number.
4. Line 345, delete “autism spectrum disorder” and parentheses around “ASD.”
Author Response
Thank you for your detailed review, and for identifying these errors. All have been address as requested with the exception of one: Line 329: we altered the text and replaced the reference with one more to the broader point of neuroplasticity. In doing so the quote was removed.